# TICAM-1/TRIF associates with Act1 and suppresses IL-17 receptor–mediated inflammatory responses

Yusuke Miyashita[1,2] , Takahisa Kouwaki[1], Hirotake Tsukamoto[1,3] , Masaaki Okamoto[1] , Kimitoshi Nakamura[2] , Hiroyuki Oshiumi[1]

**TICAM-1 (also called TRIF) is the sole adaptor of TLR3 that recognizes double-stranded RNA. Here, we report that TICAM-1 is involved not only in TLR3 signaling but also in the cytokine receptor IL-17RA signaling. We found that TICAM-1 bound to IL-17R adaptor Act1 to inhibit the interaction between IL-17RA and Act1. Interestingly, *TICAM-1* knockout promoted IL-17RA/Act1 interaction and increased IL-17A–mediated activation of NF-κB and MAP kinases, leading to enhanced expression of inflammatory cytokines and chemokines upon IL-17A stimulation. Moreover, *Ticam-1* knockout augmented IL-17A–mediated CXCL1 and CXCL2 expression in vivo, resulting in accumulation of myeloid cells. Furthermore, *Ticam-1* knockout enhanced delayed type hypersensitivity and exacerbated experimental autoimmune encephalomyelitis. *Ticam-1* knockout promoted accumulation of myeloid and lymphoid cells in the spinal cord of EAE-induced mice. Collectively, these data indicate that TICAM-1 inhibits the interaction between IL-17RA and Act1 and functions as a negative regulator in IL-17A–mediated inflammatory responses.**

## Introduction

Pattern recognition receptors (PRRs) are essential for sensing pathogens and initiating the innate immune responses. Toll-like receptors, RIG-I-like receptors, and Nod-like receptors as well as cGAS and C-type lectins play crucial roles in detecting infection of microorganisms. TLRs require several adaptor molecules to induce the innate immune response. TICAM-1 (also called TRIF), is the sole adaptor of TLR3 and triggers the signal to induce the expression of type I IFN and pro-inflammatory cytokines in response to viral dsRNA (Yamamoto et al, 2002; Oshiumi et al, 2003a). MyD88, another adaptor molecule of TLRs, is involved in the signaling of all TLRs except TLR3 (Akira & Takeda, 2004; Kawai & Akira, 2011).

Both TICAM-1 and MyD88 have a toll/interleukin-1 receptor (TIR) domain essential for TLR binding. MyD88 binds to TLRs as well as IL-1R and IL-18R via its TIR domain; thus, MyD88 is involved not only in TLR signaling but also in IL-1 and IL-18 signaling (Adachi et al, 1998; Burns et al, 1998). TICAM-1 also uses TIR domain to binds TLR3 and to induce type I IFN expression in response to TLR3 ligands (Funami et al, 2008). Previous studies have shown that TLR3 and TICAM-1 play crucial roles in antiviral innate immune responses (Wang et al, 2004; Zhang et al, 2020). Because the TIR of TICAM-1 does not bind to those of IL-1 and IL-18, TICAM-1 is believed to not be involved in cytokine receptor-mediated signaling.

In addition to TLR3, TICAM-1 is required for other PRR-mediated signaling. TICAM-1 associates with TICAM-2 (also called TRAM) and functions as an adaptor of TLR4 (Fitzgerald et al, 2003; Oshiumi et al, 2003b). cGAS is a cytoplasmic viral dsDNA sensor and requires STING to induce type I IFN expression (Chan & Gack, 2016; Chen et al, 2016; Takashima et al, 2018). STING is reported to require TICAM-1 for triggering the innate immune responses (Wang et al, 2016). In addition, TICAM-1 is involved in the DDX1, DDX21, and DHX36 protein complex to sense cytoplasmic dsRNA and induces antiviral innate immune responses (Ruan et al, 2019). TICAM-1 functions as an adaptor for the PRRs and is crucial for antiviral innate immune responses, whereas it is also reported that *Ticam-1* KO mice develop severe symptoms of experimental autoimmune encephalomyelitis (EAE), a mouse model of multiple sclerosis (Guo et al, 2008). But, it remains unclear whether TICAM-1 is directly involved in IL-17 signaling.

IL-17A plays a key role in EAE, delayed type hypersensitivity (DTH), collagen-induced arthritis, and ulcerative colitis (Nakae et al, 2002; Ishigame et al, 2009; Sarkar et al, 2014; Nanki et al, 2020). IL-17A induces the expression of cytokines and chemokines, such as IL-6, CXCL1, and CXCL2, in macrophages and epithelial cells (McGeachy et al, 2019). IL-17A is an agonist of IL-17 receptor comprising IL-17RA and IL-17RC (Chang & Dong, 2011; Li et al, 2019). IL-17RA are expressed in many types of cells, including fibroblasts, macrophages, and epithelial cells, and plays a crucial role in IL-17A–mediated cytokine and chemokine expression (Chang & Dong, 2011; Hu et al, 2011; Li et al, 2019). IL-17RA interacts with an adaptor molecule Act1 (Chang & Dong, 2011; Hu et al, 2011; Li et al, 2019).

[1]Department of Immunology, Graduate School of Medical Sciences, Faculty of Life Sciences, Kumamoto University, 1-1-1 Honjo, Kumamoto, Japan [2]Department of Pediatrics, Graduate School of Medical Sciences, Kumamoto University, 1-1-1 Honjo, Kumamoto, Japan [3]Division of Clinical Immunology and Cancer Immunotherapy, Center for Cancer Immunotherapy and Immunobiology, Graduate School of Medicine, Kyoto University, Kyoto, Japan

Correspondence: oshiumi@kumamoto-u.ac.jp; tsukamoto.hirotake.4j@kyoto-u.ac.jp

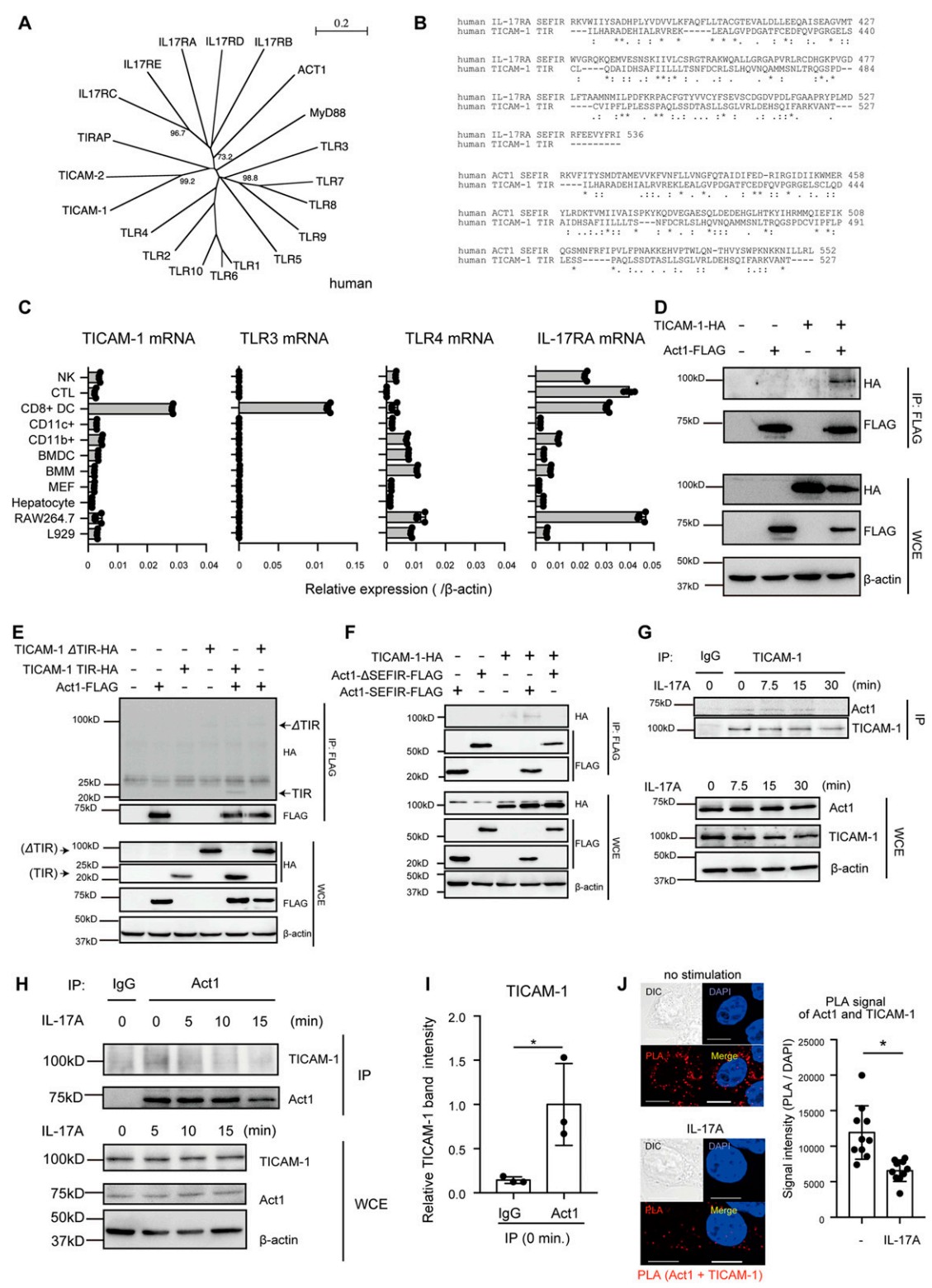

**Figure 1. Act1 interacts with TICAM-1.**
**(A)** Phylogenetic tree of SEFIR and TIR domains of IL-17 receptors, TLRs, and their adaptor molecules. **(B)** Multiple amino acid sequence alignment of SEFIR or TIR domain in IL-17RA, Act1, and TICAM-1 with Clustal W. ("*" indicates identical, ":" indicates strong similarity, and "." indicates weak similarity). **(C)** mRNA levels of TICAM-1, TLR3, TLR4, and IL-17RA in mouse primary cells and several cell lines were determined through RT-qPCR and normalized to β-actin. **(D, E, F)** Whole cell extracts (WCEs) were prepared from HEK293FT cells expressing Act1-SEFIR-FLAG, Act1-ΔSEFIR-FLAG, TICAM-1-HA, TICAM-1-TIR-HA, and TICAM-1-ΔTIR-HA as indicated. Immunoprecipitation (IP) was performed with the anti-FLAG antibody. **(G)** HeLa cells were either left untreated or treated with 100 ng/ml IL-17A. WCEs were immunoprecipitated with control IgG and

IL-17RA and Act1 have a common domain called the similar expression to fibroblast growth factor genes and IL-17R (SEFIR) domain, which is essential for the interaction between IL-17RA and Act1 (Qian et al, 2007). The SEFIR domain is known to be similar and related to the TIR domain, and both domains belong to a STIR domain super family (Novatchkova et al, 2003). Because TICAM-1, Act1, and IL-17RA share structurally similar domains, we investigated whether TICAM-1 is involved in IL-17 signaling and found that TICAM-1 plays a crucial role in IL-17–mediated signaling. Our study elucidates a cross talk between TLRs and IL-17Rs via TICAM-1 in the inflammatory responses.

# Results

### Act1 physically interacts with the TICAM-1 adaptor

TICAM-1, IL-17RA, and Act1 contain a TIR or SEFIR domain (Fig S1A). We compared amino acid sequences of SEFIR and TIR domains of IL-17Rs and TLRs. A phylogenetic tree implied that the node of Act1 was relatively close to that of TICAM-1, TICAM-2, and TIRAP (Fig 1A), and alignment of SEFIR and TIR domains indicated similarities among the TIR and SEFIR domains (Fig 1B), as previously reported (Novatchkova et al, 2003; Mellett et al, 2015). We compared the expression profile of TICAM-1 to those of TLRs and IL-17RA. TLR4 and IL-17RA were expressed in various cell types, whereas TLR3 was specifically expressed in CD8$^+$ DCs (Fig 1C). This is consistent with previous observation that TLR3 is highly expressed in CD8$^+$ DCs (Schulz et al, 2005). Although TICAM-1 expression level was high in CD8$^+$ DCs, it was also expressed in other cell types (Fig 1C), implying the involvement of TICAM-1 in TLR3-independent pathways. Microarray data in Reference Database of Immune cells also show that the expression profiles of TLR3 and TLR4 are different from that of TICAM-1 (Fig S1B).

Because TIR and SEFIR domains are protein–protein interaction domains, we investigated whether TICAM-1 binds to Act1. As expected, HA-tagged TICAM-1 was co-immunoprecipitated with FLAG-tagged Act1 (Fig 1D). HA-tagged TIR domain of TICAM-1 (TICAM-1-TIR-HA), but not HA-tagged TICAM-1 without TIR domain (TICAM-1-ΔTIR-HA), was co-immunoprecipitated with FLAG-tagged Act1 (Fig 1E). In addition, deletion of SEFIR domain of Act1 abrogated the interaction between TICAM-1 and Act1 (Fig 1F). These data suggest that the TIR and SEFIR domains are important for the physical interaction. Next, we investigated the physical interaction of endogenous proteins in cells stimulated with IL-17A. Endogenous Act1 was co-immunoprecipitated with endogenous TICAM-1 in the presence and absence of IL-17A stimulation (Figs 1G and S1C). Endogenous TICAM-1 was also co-immunoprecipitated with endogenous Act1 (Fig 1H and I). These physical interactions between TICAM-1 and

Act1 were a little reduced at later time points after IL-17A stimulation (Fig 1G and H) (see the Discussion section). To further confirm the physical interaction between TICAM-1 and Act1, we performed a proximity ligation assay (PLA), which detects the colocalization of two proteins. As expected, the PLA signals showing the colocalization of FLAG-tagged Act1 with HA-tagged TICAM-1 were detected in the cytoplasm (Fig S1D). Moreover, the PLA signals of colocalization between endogenous TICAM-1 and Act1 were also detected in the cytoplasm, and the number of signals was reduced after IL-17A stimulation (Fig 1J). These data indicate that TICAM-1 binds to Act1 in resting and IL-17A-stimulated cells.

### TICAM-1 modulates the expression of cytokines and chemokines in response to IL-17A

Next, we investigated the effect of TICAM-1 on cytokine expression in response to IL-17A, and found that the *TICAM-1* siRNA knockdown enhanced IL-17–mediated CXCL1 mRNA expression in HeLa cells (Fig S2A). To further test the effect of TICAM-1, we generated *TICAM-1* KO HeLa cells using the CRISPR-Cas9 system and confirmed TICAM-1 expression by Western blotting (Fig 2A). As described previously (Oshiumi et al, 2003a), *TICAM-1* KO markedly reduced the expression of IFN-β mRNA induced by TLR3 stimulation with short poly I:C (Fig 2B). Interestingly, IL-17A–induced CXCL1 and CXCL2 mRNA expression was significantly increased by *TICAM-1* KO (Fig 2C). Another *TICAM-1* KO clone (KO2) also exhibited enhanced CXCL1 mRNA expression upon IL-17A simulation (Fig 2D). In addition, *Ticam-1* siRNA knockdown augmented IL-17A–mediated CXCL1 and IL-6 mRNA expression in L929 and a macrophage-like cell line RAW264.7 (Fig 2E and F). To test the effect of TICAM-1 in the IL-17A signaling in primary cells, we prepared MEFs, and investigated the effect of TICAM-1 on IL-17A–mediated chemokine expression. *Ticam-1* KO enhanced CXCL1 and CXCL2 mRNA expression in MEFs (Fig 2G). Collectively, these data suggest that TICAM-1 suppresses IL-17–mediated pro-inflammatory cytokine and chemokine expression.

Because previous studies have reported that Act1 directly stabilizes CXCL1 mRNA, thereby increasing the mRNA level (Herjan et al, 2018), we investigated whether TICAM-1 is involved in Act1-mediated stabilization of the mRNA. Transcriptions were inhibited by actinomycin D treatment, and the degradation of RNAs was determined by RT-qPCR. However, there were no significant differences in the degradation of CXCL1 mRNA in IL-17A–treated and untreated cells (Fig S2B).

### TICAM-1 suppresses IL-17RA–mediated signaling

Because the NF-κB pathway is activated through IL-17A stimulation (Wu et al, 2015), we assessed the role of TICAM-1 in IL-17A–mediated NF-κB activation. Phosphorylation of the IκB and NF-κB subunit p65

anti–TICAM-1 antibody. The proteins were subjected to SDS–PAGE and detected by Western blotting with indicated antibodies. The data were representative of three independent experiments. **(H, I)** HeLa cells were either left untreated or treated with 100 ng/ml IL-17A. **(H)** WCEs were immunoprecipitated with control IgG and anti-Act1 antibody (H). The data were representative of three independent experiments. The band intensities of immunoprecipitated TICAM-1 from unstimulated cells were measured and normalized to the average band intensity of immunoprecipitated TICAM-1 of samples immunoprecipitated with anti-Act1 antibody. **(I)** The results show ±SD means (n = 3 *P < 0.05; *t* test) (I). **(J)** HeLa cells were stimulated with 0 or 100 ng/ml of IL-17A for 15 min, and then proximity ligation assay was performed with anti-Act1 and anti–TICAM-1 antibodies (left panel). The proximity ligation assay signal intensities were normalized to those of DAPI (right panel). The results show means ± SD (*P < 0.05; *t* test). The white bar represents 10 μm.

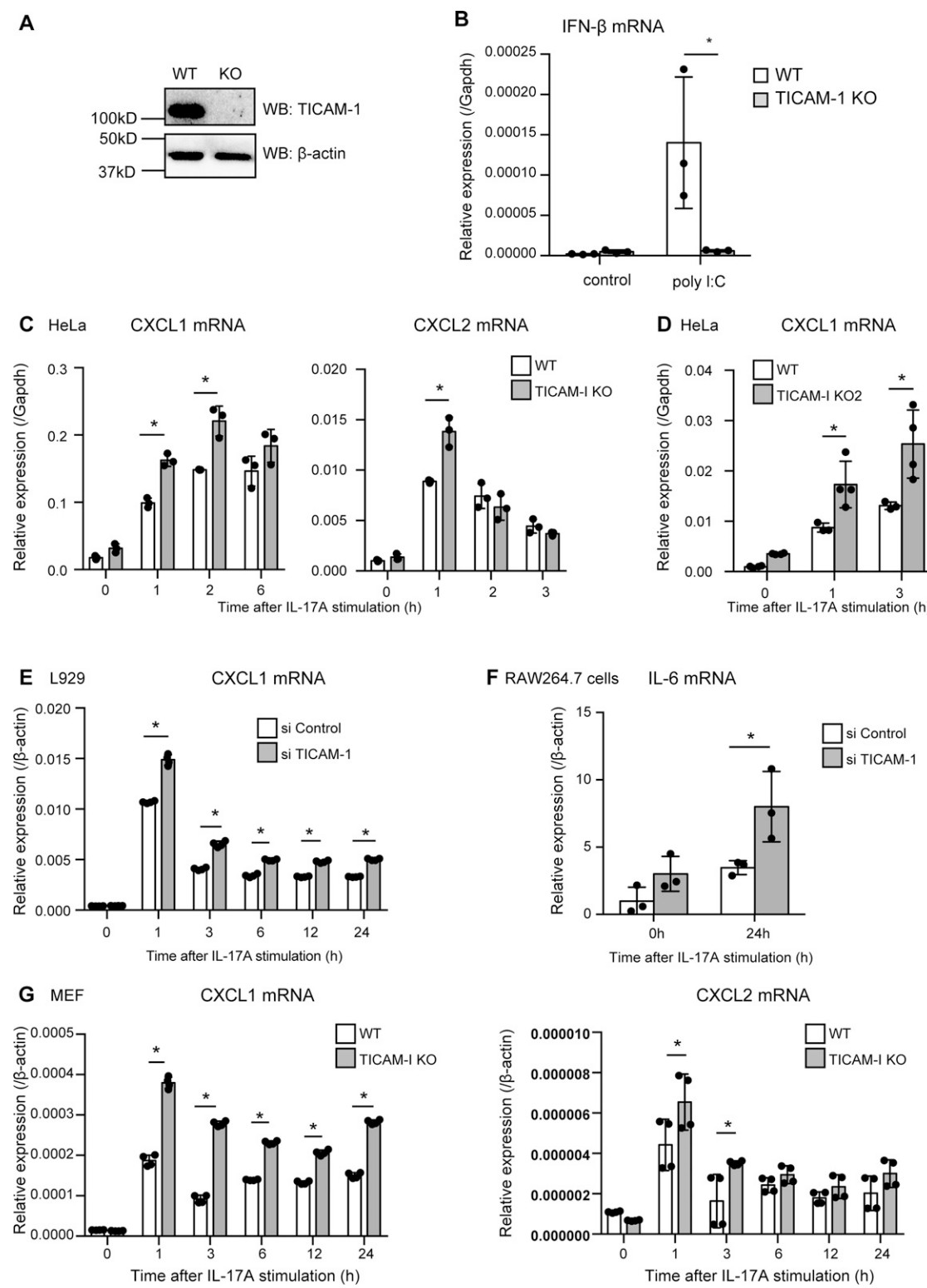

**Figure 2. TICAM-1 suppresses IL-17–mediated expression of inflammatory cytokines and chemokines.**
**(A)** Expression of TICAM-1 in WT and *TICAM-1* KO HeLa cells was analyzed by Western blotting with anti-$\beta$ actin and anti–TICAM-1 antibodies. **(B)** IFN-$\beta$ mRNA levels in WT and *TICAM-1* KO HeLa cells upon poly I:C stimulation was determined using RT-qPCR. The results show ± SD means (n = 3 *P < 0.05; *t* test). **(C, D)** WT and *TICAM-1* KO HeLa cells were stimulated with 50 ng/ml IL-17A, and the expression of CXCL1 and CXCL2 mRNA was determined by RT-qPCR. The results show ± SD means (n = 3 *P < 0.05; *t* test). **(D)** Another clone (KO2) was used in panel (D). The results show ± SD means (n = 3–4 *P < 0.05; *t* test). **(E)** *Ticam-1* or negative control siRNA were transfected into L929 cells, which were then stimulated with IL-17A. The expression of Cxcl1 mRNA was determined by RT-qPCR. The results show ± SD means (n = 4 *P < 0.05; two-way ANOVA). **(F)** *Ticam-1* or negative control siRNA were transfected into RAW264.7 cells, which were then stimulated with IL-17A. The expression of IL-6 mRNA was determined by RT-qPCR. The results show ± SD means (n = 3 *P < 0.05; *t* test). **(G)** WT or *Ticam-1* KO MEFs were stimulated with IL-17A, and the expression of CXCL1 and CXCL2 mRNA was determined by RT-qPCR. (n = 4 *P < 0.05; two-way ANOVA).

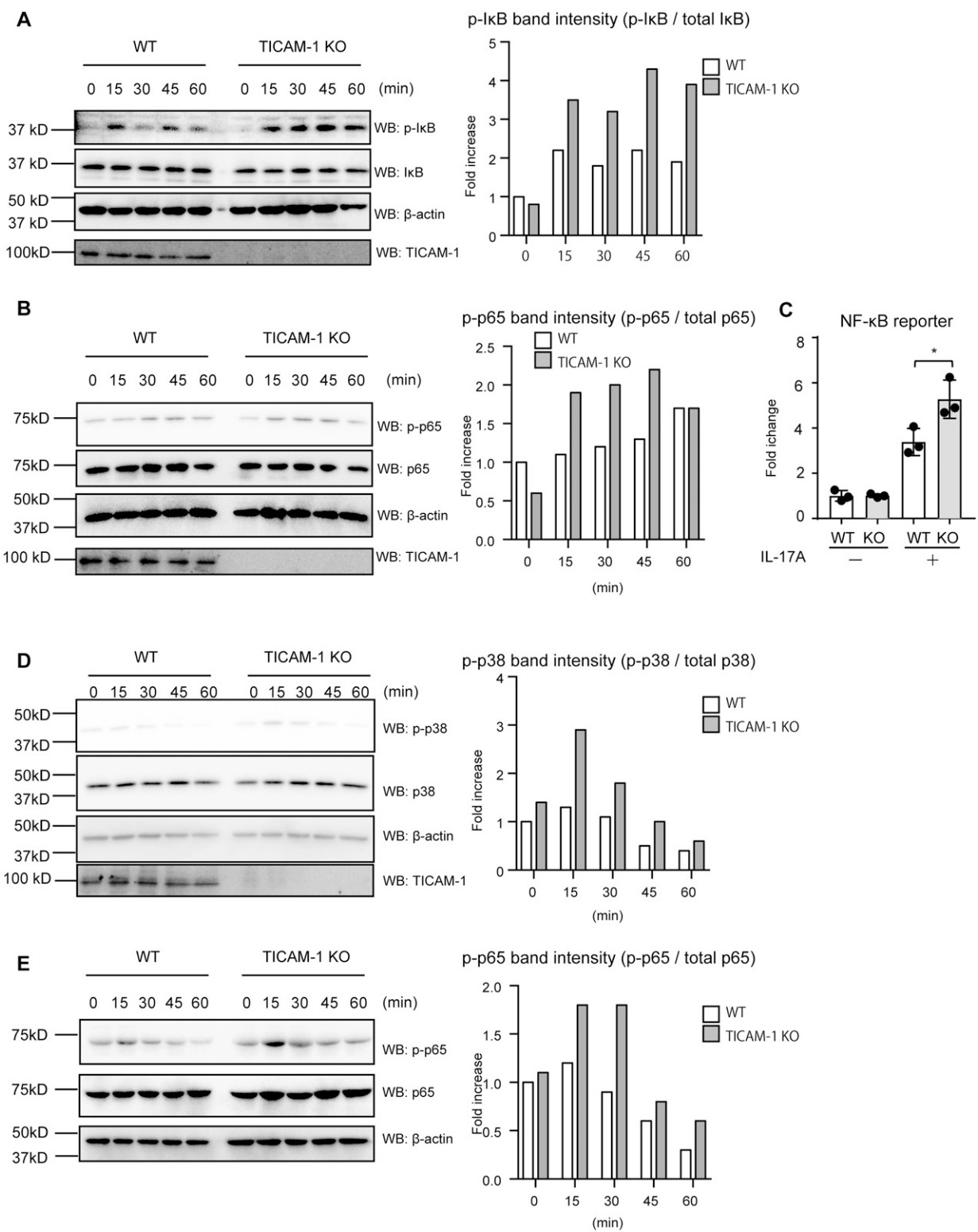

**Figure 3. TICAM-1 regulates IL-17A–mediated activation of transcription factors.**
**(A, B)** WT or *TICAM-1* KO HeLa cells were stimulated with IL-17A (50 ng/ml) for the indicated time, and whole cell extracts (WCEs) were blotted with anti-phospho-IκB (p-IκB), IκB, TICAM-1, and β-actin (A) and anti-phospho-p65 (p-p65), p65, TICAM-1. and β-actin (B) antibodies as indicated. Each band intensity of p-IκB and p-p65 was measured and normalized to that of total IκB and p65, respectively. Fold increases were calculated by dividing the normalized intensity at each time point with that at 0 min in WT HeLa cells. The data were representative of two independent experiments. **(C)** WT and *TICAM-1* KO HeLa cells were transfected with a reporter plasmid containing the NF-κB promoter and *Renilla* luciferase vector (internal control). After 18 h, cells were stimulated with 50 ng/ml IL-17A. NF-κB reporter activities were

is a hallmark of NF-κB pathway activation (Karin & Ben-Neriah, 2000). IL-17A stimulation induced IκB and p65 phosphorylation (p-IκB and p-p65), which was augmented by *TICAM-1* KO in HeLa cells (Fig 3A and B). Moreover, reporter gene analysis revealed that IL-17A–induced NF-κB reporter activity was significantly augmented by *TICAM-1* KO (Fig 3C), suggesting that TICAM-1 suppresses the activation of NF-κB in response to IL-17A. MAP kinases, such as p38, are activated by IL-17A stimulation (Hata et al, 2002), and we found that p38 phosphorylation (p-p38) markedly increased by *TICAM-1* KO in HeLa cells (Fig 3D).

Next, we compared NF-κB activation in primary cells. *Ticam-1* KO increased IL-17A–mediated p-p65 levels in MEFs (Fig 3E). Collectively, these data indicate that TICAM-1 suppresses the activation of transcription factors downstream of IL-17RA, which is consistent with the notion that TICAM-1 suppresses IL-17RA–mediated cytokine expression.

Next, we investigated the post-translational modifications of TICAM-1 after IL-17A stimulation. First, we tried to detect poly-ubiquitin chains, but we could not detect polyubiquitination of TICAM-1 after IL-17A stimulation at least in our experimental condition (Fig S3A). Second, we investigated the phosphorylation of TICAM-1 after IL-17A stimulation. To detect the phosphorylation, we used Phos-tag gel, in which phosphorylated proteins migrate more slowly than unphosphorylated proteins. However, we could not detect any delayed migration of TICAM-1 bands after IL-17A stimulation (Fig S3B).

### TICAM-1 disrupts the interaction between IL-17RA and Act1

Next, we investigated the underlying mechanism of how TICAM-1 inhibited IL-17RA signaling. We hypothesized that TICAM-1 would disrupt the IL-17RA–Act1 interaction. To test this hypothesis, we performed immunoprecipitation assays. Myc-tagged IL-17RA was co-immunoprecipitated with FLAG-tagged Act1, and TICAM-1 over-expression reduced the interaction between Act1 and IL-17RA in HEK293FT cells (Fig 4A). We repeated the experiments and confirmed that the reduction of the interaction was statistically significant (Fig 4B). Next, endogenous IL-17RA was co-immunoprecipitated with endogenous Act1 in the presence and absence of IL-17A, and TICAM-1 overexpression inhibited the physical interaction between IL-17RA and Act1 in HeLa cells (Fig 4C). Next, we investigated whether the TIR domain of TICAM-1 was responsible for the inhibition. Expression of the TICAM-1 TIR fragment inhibited the interaction between ectopically expressed Act1 and IL-17RA (Fig S4). Moreover, the expression of the TIR domain inhibited the interaction between endogenous IL-17RA and Act1 upon IL-17A stimulation (Fig 4D), suggesting that the TIR domain is responsible for the inhibition. Conversely, *TICAM-1* KO increased the interaction between endogenous Act1 and IL-17RA in the presence of IL-17A (Fig 4E).

We conducted a PLA to investigate whether TICAM-1 influence the colocalization of IL-17RA with Act1 upon IL-17A stimulation. The number of the PLA signals of IL-17RA and Act1 increased after IL-17A stimulation in WT HeLa cells (Fig 4F), and *TICAM-1* KO further increased the number of the PLA signals upon IL-17A stimulation in HeLa cells (Fig 4G and H). These data indicate that TICAM-1 interferes with the physical interaction between Act1 and IL-17RA. This model is consistent with our observation that *TICAM-1* KO enhanced IL-17RA–mediated cytokine expression.

### TICAM-1 deficiency promotes IL-17–induced inflammatory response in vivo

Our in vitro results indicate that TICAM-1 suppresses IL-17A–mediated expression of cytokines and chemokines, such as CXCL1 and CXCL2, which play crucial roles in recruiting myeloid cells, such as CD11b⁺ Gr1⁺ cells (Acharyya et al, 2012; Girolomoni et al, 2012). Next, we assessed the physiological significance of TICAM-1–mediated inhibition of IL-17A signaling in vivo. We injected recombinant IL-17A intraperitoneally into WT and *Ticam-1* KO mice and evaluated the accumulation of inflammatory cells in the peritoneal cavity. Absolute number of total cells increased in the peritoneal cavity upon IL-17A stimulation, and the accumulation was significantly augmented by TICAM-1 KO (Fig 5A). Moreover, *Ticam-1* KO mice displayed excessive accumulation of Gr-1⁺ CD11b⁺ neutrophils in the peritoneal cavity (Fig 5B–D) as well as Gr-1⁻ CD11bᵐⁱᵈᵈˡᵉ CD11c⁻ F4/80⁻ monocytes (Fig 5B and E) after peritoneal injection of recombinant IL-17A compared to WT. Furthermore, *Ticam-1* KO mice intranasally injected with IL-17A also displayed excessive accumulation of Gr-1⁺ CD11b⁺ neutrophils in the lung (Fig 5F and G). We confirmed that *Ticam-1* KO increased mRNA expression of CXCL1, CXCL2, and IL-6 in the lung in response to IL-17A administration (Fig 5H and I). Considering that TICAM-1 augmented the inflammatory response to IL-17A without TLR ligands, our data indicate a crucial role of TICAM-1 in regulating IL-17A–mediated immune response in vivo.

### TICAM-1 attenuates DTH and autoimmune encephalomyelitis

Next, we further assessed physiological significance of TICAM-1 in IL-17 signaling in vivo. IL-17 plays a critical role in the pathogenesis of DTH and autoimmune diseases, such as EAE (Nakae et al, 2002; McGinley et al, 2020). Because TICAM-1 suppressed IL-17A–mediated inflammatory responses, we investigated whether *Ticam-1* KO enhances DTH and exacerbates EAE pathogenesis. To observe DTH, we performed contact hypersensitivity assay using 1-fluoro-2,4-dinitrobenzene (DNFB) and found that DNFB-induced ear swelling was significantly enhanced in *Ticam-1* KO mice (Fig 6A).

---

determined 6 h after IL-17A stimulation. The results show ± SD means (n = 3 *P < 0.05; *t* test). **(D)** WT or *TICAM-1* KO HeLa cells were stimulated with IL-17A (50 ng/ml) for the indicated time, and WCEs were blotted with anti-phospho-p38 (p-p38), p38, TICAM-1, and β-actin antibodies as indicated. Each band intensity of phosphor-p38 (p-p38) was measured and normalized to that of total p38. Fold increases were calculated by dividing the normalized intensity at each time point with that at 0 min in WT HeLa cells. The data were representative of two independent experiments. **(E)** WT and *Ticam-1* KO MEFs (E14) were stimulated with 50 ng/ml IL-17A for the indicated time. WCEs were blotted with anti-p-p65, p65, and β-actin antibodies. Each band intensity of p-p65 was measured and normalized to that of total p65. Fold increases were calculated by dividing the normalized intensity at each time point with that at 0 min in WT MEFs. The data were representative of two independent experiments.

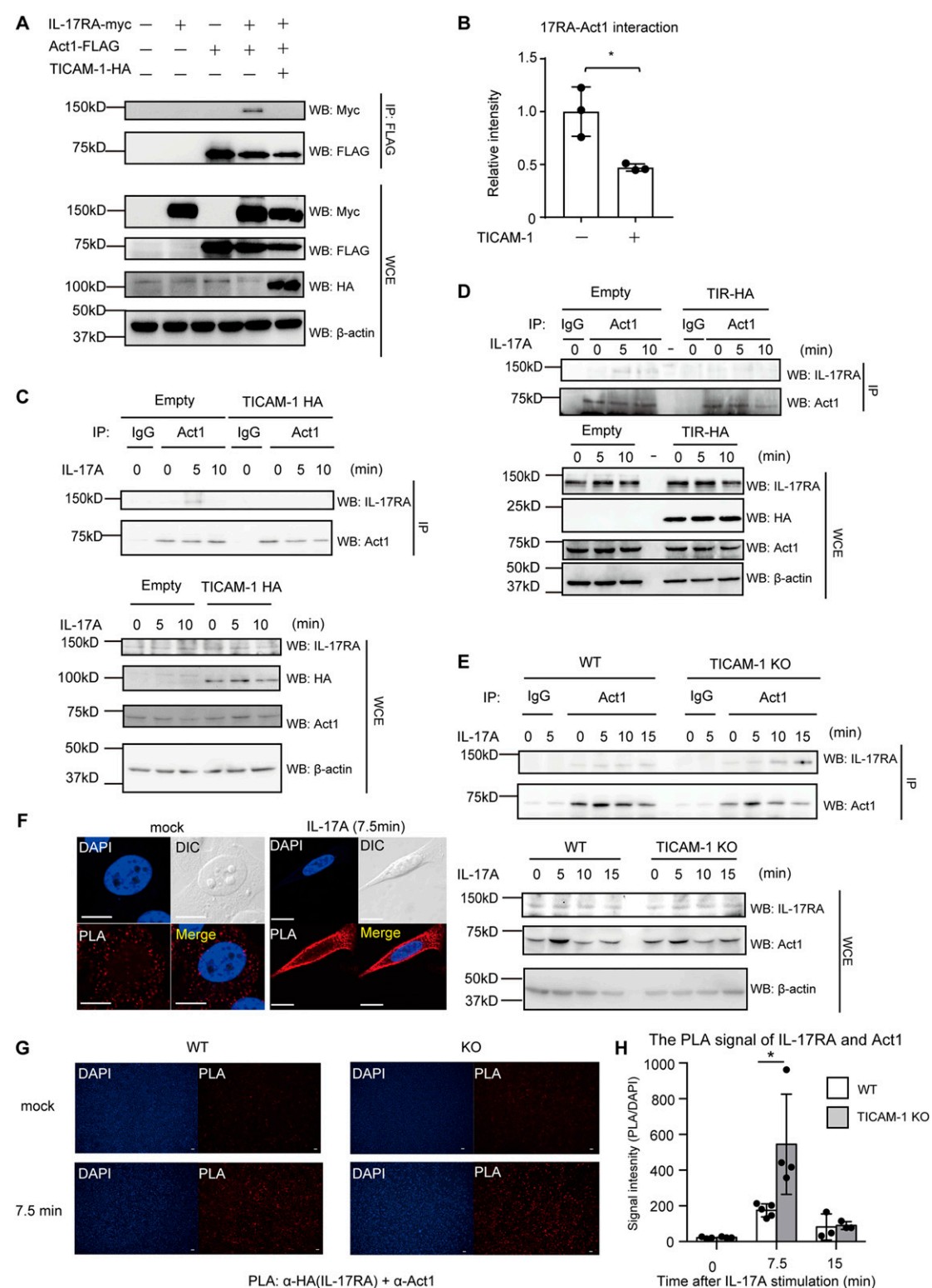

**Figure 4. TICAM-1 inhibits the interaction between IL-17RA and Act1.**
**(A, B)** Whole cell extracts were prepared from HEK293FT cells expressing Act1-FLAG, IL-17RA-myc, and TICAM-1-HA. Immunoprecipitation was performed with the anti-FLAG antibody. The data were representative of three independent experiments. **(B)** The band intensities of immunoprecipitated IL-17RA-myc were measured and normalized to the average band intensity of immunoprecipitated IL-17RA-myc of samples co-expressed IL-17RA-myc and Act1-FLAG (B). The results show ± SD means (n = 3 *P < 0.05; t test). **(C, D)** HeLa cells transfected with empty, TICAM-1-HA (C), or TICAM-1-TIR-HA (D) expression vector were either left untreated or treated with 50 ng/ml IL-17A for 5- or 10-min. Immunoprecipitation was performed with the anti-Act1 antibody and control IgG. **(E)** WT or *TICAM-1* KO HeLa cells were stimulated with IL-17A for the

Next, we conducted mouse EAE assay. Clinical symptoms of EAE were exacerbated by *Ticam-1* KO (Fig 6B). Absolute number of total cells in draining lymph nodes increased by *Ticam-1* KO (Fig 6C), and spleen size was enlarged (Fig 6D) in EAE-induced *Ticam-1* KO mice, suggesting that the inflammatory response is augmented in *Ticam-1* KO mice during EAE pathogenesis.

Considering that TICAM-1 suppressed IL-17A–mediated chemokine expression, we expected that *Ticam-1* KO would increase the accumulation of myeloid and lymphoid cells in EAE-induced mice. Flow cytometry analysis of the spinal cord demonstrated that the percentage of central nervous system-infiltrating CD4$^+$ and Gr-1$^-$ F4/80$^+$ cells significantly increased in *Ticam-1* KO mice compared with WT mice at 19 d post EAE induction (Fig 6E and F). Next, we investigated whether the severity of EAE in *Ticam-1* KO mouse is dependent on IL-17RA–mediated signaling. We intraperitoneally injected anti-IL-17A blocking antibody during the induction of EAE and found that the blocking of IL-17A–mediated signaling inhibited EAE symptoms in both of WT and *Ticam-1* KO mice (Fig 6G), suggesting that EAE was induced in an IL-17A–dependent manner both in WT and *Ticam-1* KO mice in the experimental condition. Collectively, our data indicate that TICAM-1 is involved in IL-17A–mediated immune responses in vivo.

# Discussion

TICAM-1 has been reported to be the solo adaptor molecule of TLR3, and in this study, we elucidated another role of TICAM-1 in IL-17A–mediated signaling. Previously, Mellett et al have reported a cross talk between TLRs and IL-17Rs, in which the SEFIR domain of IL-17RD inhibits TLR signaling (Mellett et al, 2015). Our findings provide another mechanism underlying the cross talk between TLRs and IL-17Rs. It is expected that the difference in TICAM-1 expression levels would make a difference in the cytokine expression profile after stimulations with TLR3 and/or IL-17RA ligands.

TIR domain resembles the SEFIR domain in structure (Maitra et al, 2007). Since TIR and SEFIR domains are responsible for protein–protein interactions, we hypothesized that TICAM-1 would associate with SEFIR domain containing proteins and found the interaction between TICAM-1 and Act1. Although TICAM-1 promotes TLR3-mediated type I IFN expression, TICAM-1 inhibited the Act1–IL-17RA interaction, thereby attenuating IL-17A–mediated cytokine and chemokine expression. Thus, expression levels of TICAM-1 are expected to determine the strengths of TLR3 and IL-17RA signaling. In addition, we found that IL-17A stimulation reduced the interaction between TICAM-1 and Act1. This is correlated with previous model that IL-17A induces the interaction between IL-17RA and Act1 (Qian et al, 2007).

It is still unclear how TICAM-1 inhibits the interaction between IL-17RA and Act1. It is possible that TICAM-1 directly disrupt the physical interaction between IL-17RA and Act1. Another possibility is that TICAM-1 sequestrates Act1, thereby keeping Act1 away from IL-17RA. In our experimental conditions, Act1 bound to IL-17RA immediately after IL-17A stimulation, whereas TICAM-1 dissociated from Act1 at later time points. Thus, dissociation of TICAM-1 from Act1 would not be a prerequisite for the binding of Act1 to IL-17RA, but it might modulate the binding stability of Act1 to IL-17RA. Another possibility is that other factors, such as TICAM-2/TRAM, affected their dissociation and/or associations. It is also possible that post-translational modification affected the TICAM-1–mediated inhibition. Some experimental conditions, such as transfection of plasmids, might affect the sensitivity of cells to IL-17A. Further studies are required to fully reveal the underlying mechanism.

IL-17A triggers the expression of inflammatory responses, such as chemokine expression, leading to accumulation of myeloid and lymphoid cells. Because TICAM-1 attenuated the interaction between IL-17RA and Act1, it was expected that *Ticam-1* KO would accelerate recruitment and accumulation of the immune cells. Indeed, *Ticam-1* KO increased accumulation and/or recruitment of myeloid cells. These observations indicate an important in vivo role of TICAM-1–mediated inhibition of the interaction in ameliorating excessive IL-17A–mediated inflammatory responses.

IL-17A are important for hyper sensitivities and autoimmune diseases, and it has been reported that IL-17A promotes mouse EAE (Ogura et al, 2008). Our study showed that *Ticam-1* KO exacerbated mouse EAE. It is still possible that *Ticam-1* KO affected mouse EAE via TICAM-1–dependent TLR responses. For instance, a previous study has shown that TICAM-1–dependent type I IFN expression promotes the production of anti-inflammatory cytokine IL-27 that attenuates IL-17 production, thereby ameliorating mouse EAE (Guo et al, 2008). However, we prefer the interpretation that not only TICAM-1–dependent type I IFN production but also TICAM-1–mediated inhibition of IL-17RA and Act1 interaction ameliorate mouse EAE. This is because our in vitro studies showed that TICAM-1 inhibited the interaction between Act1 and IL-17RA even in the absence of any TLR simulation. Moreover, IL-17A–induced myeloid cell accumulation in vivo was enhanced by *Ticam-1* KO in the absence of TLR stimulation. Therefore, we expect that TICAM-1 would directly and indirectly inhibit IL-17RA signaling.

Several viral infections reportedly trigger the development of IL-17–mediated autoimmune disease (Getts et al, 2013). Because several viruses target TICAM-1 to infect host cells, our findings imply that viruses might affect not only TLR but also IL-17 signaling pathways. IL-17RA signaling is also involved in the pathogenesis of acute respiratory distress syndrome (ARDS) caused by viral infection (Crowe et al, 2009). Cytokine release syndrome in COVID-19 patients sometimes results in ARDS because of pulmonary edema

---

indicated time. Immunoprecipitation was performed with control IgG or anti-Act1 antibody. **(F, G, H)** HeLa cells were transfected with IL-17RA-HA expression vector for 24 h, then stimulated with 100 ng/ml IL-17A for indicated time. Proximity ligation assay (PLA) was performed using anti-HA and anti-Act1 antibodies. **(F)** The PLA signals in mock or stimulated WT cells were shown in panel (F). **(G)** The PLA signals in WT and *TICAM-1* KO HeLa cells upon IL-17A stimulation were shown in panel (G). The PLA signal intensities were normalized to the DAPI numbers in WT and *TICAM-1* KO HeLa cells. The results show ± SD means (n = 3–6 *P < 0.05; *t* test). **(F, G)** The white bar represents 10 (F) and 100 *μm* (G).

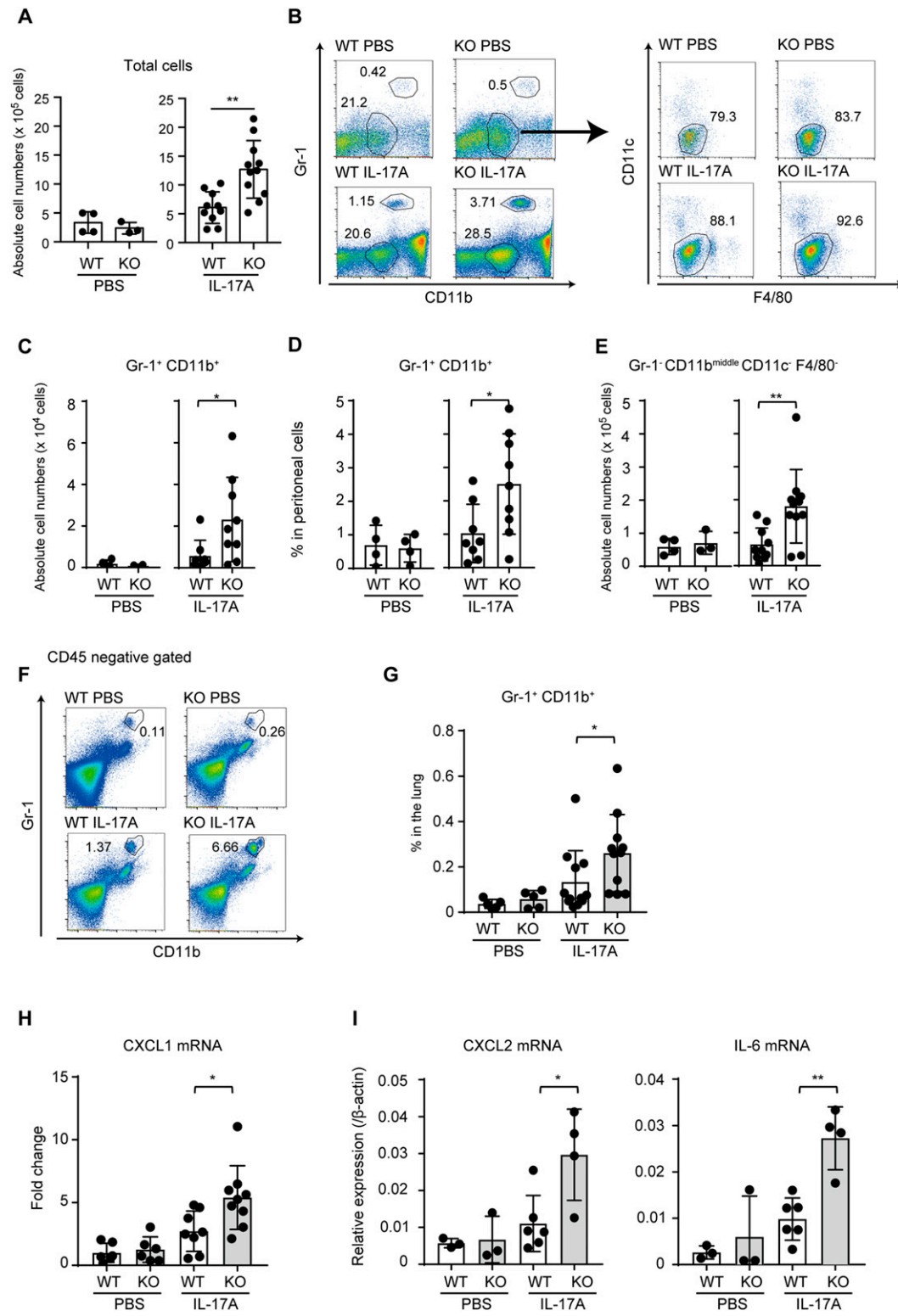

**Figure 5. Physiological significance of TICAM-1 in IL-17–induced cytokine expression in vivo.**
**(A)** WT and *Ticam-1* KO male mice were intraperitoneally injected with PBS (WT n = 4, KO n = 3) or 0.5 µg IL-17A (WT n = 7, KO n = 8). 5 h after injection, cells were collected. Total cell number in peritoneal cavity is shown. **(B, C, D)** Representative plots (B), absolute numbers (C), and percentages of Gr-1⁺ CD11b⁺ neutrophils (D) from male mice treated in panel A (PBS: WT n = 4, KO n = 4. IL-17A: WT n = 8, KO n = 9). **(A, E)** Absolute numbers of Gr-1⁻ CD11b^middle CD11c⁻ F4/80⁻ monocytes from male mice treated as in panel (A) (PBS: WT n = 4, KO n = 3. IL-17A: WT n = 10, KO n = 11). **(F, G)** WT and *Ticam-1* KO female mice were intranasally injected with PBS or IL-17A (2 µg). 5 h after injection, the lung was excised, and cells were prepared. Intravascular CD45 staining was performed just before mice were euthanized. **(F, G)** Representative plots (F) and

and lung failure. Several groups reported that peripheral Th17 cells and IL-17A in serum are higher in COVID-19 patients than healthy donors (Xu et al, 2020; Ghazavi et al, 2021). PRRs and these adaptor molecules are also involved in ARDS caused by viral infection, and it has been reported that TICAM-1 had a protective role in ARDS caused by SARS-CoV, showing increased mortality, weight loss, and reduced lung function in *Ticam-1* KO mice (Totura et al, 2015). Further studies are required to assess physiological significance of TICAM-1–mediated regulation of IL-17A signaling in ARDS. Recently, IL-17–dependent immune responses have been intensively studied in viral infectious diseases (Parrot et al, 2020; Pierce et al, 2020). Thus, the cross talk between TLR3 and IL-17A signaling via TICAM-1 is expected to be important to understand the pathogenesis of those serious infectious diseases, including recent coronavirus disease 2019.

# Materials and Methods

## Reagent and mice

*Ticam-1/Trif* KO C57BL/6 mice were obtained from Oriental Bio Service. C57BL/6JJmsSlc mice were purchased from Japan SLC, Inc. All the experimental procedures were approved by the Institutional Animal Committee of Kumamoto University and performed in accordance with the guidelines. Recombinant mouse IL-17A was purchased from R&D System, and human IL-17A was purchased from TOYOBO. Recombinant TNF-$\alpha$ was purchased from PEPRO-TECH. Actinomycin D was from Sigma-Aldrich. Antibodies against Gr-1 (eBioscience), CD45.2, F4/80(clone BM8), CD11b, Ly6c, CD11c (BioLegend), and CD4 (TONBO) were used for cytometric analysis. Antibodies against TRIF (#4596), p-p65 (Ser536) (93H1), p65 (D14E12), p-I$\kappa$B (Ser32) (14D4), I$\kappa$B (44D4), p-p38 (Thr180/182) (D3F9), p38 (D13E1) (Cell Signaling Technology), $\beta$-actin (MBL), IL-17R (clone G-9), Act1 (clone WW-18) (Santa Cruz), myc (clone 4A6) (Millipore) HA (#H6908), and FLAG (#F3165) (Sigma-Aldrich) were used for Western blot analysis.

## Contact hypersensitivity assay and in vivo IL-17 stimulation

Age- (8–12 wk) and sex-matched mice were sensitized by applying 50 $\mu$l of freshly prepared 0.5% of 1-fluoro-2,4-dinitrobenzene (DNFB) in acetone: olive oil (4:1) to the shaved abdomen. 4 d later the right ears were challenged with 10 $\mu$l of freshly prepared 1% of DNFB. Right and left ear thickness before and 24 h after the challenge was measured. The following formula was used for calculating ear swelling (%). % Ear swelling = (right ear thickness/left ear thickness) × 100. For in vivo IL-17 stimulation, mouse recombinant IL-17A (0.5 or 5 $\mu$g) or vehicle control (PBS) was intraperitoneally injected or intranasally inoculated into the mice, and 5 h later peritoneal

exudate cells or lung tissues were harvested for further analysis, respectively.

## EAE induction

EAE induction was previously described. Briefly, age- (8–12 wk) and sex-matched mice were immunized with 200 $\mu$g MOG$_{35-55}$ peptide (Scrum) and complete Freund's adjuvant containing 5 mg/ml (0.5 mg/mouse) heat-killed *Mycobacterium tuberculosis* (BD). Mice were injected with this emulsion at the tail base. Pertussis toxin (500 ng/mouse) was injected intraperitoneally on days 0 and 2. For IL-17A neutralization study, mice were treated intraperitoneally with 100 $\mu$g of anti-IL-17A (clone 17F3; BioXCell) or mouse IgG$_1$ isotype control antibody (clone MOPC-21; BioXCell) every 3 d, starting at day −1 of EAE immunization until day 14. Immunized mice were assessed everyday by clinical scores as follows: no clinical signs, 0; partially limp tail, 0.5; completely limp tail, 1; hind leg inhibition (hind legs fall when mice are dropped on a wire rack), 1.5; dragging of one hind leg, 2; dragging of both hind legs or paralysis of one hind leg, 2.5; paralysis of both hind legs, 3; paralysis of both hind legs and hind legs are together on one side of the body, 3.5; limp tail, complete hind legs, and mouse is minimally moving around the cage, 4; mouse is found dead due to paralysis, 5.

## Primary cells and cell lines

TICAM-1 KO cells were generated using the CRISPR/Cas9 system (Ran et al, 2013). Guide RNA sequence for *TICAM-1* was 5′-CACC-GAGTCCGAAACACCGTCAATG-3′. This guide RNA was cloned into a BbsI site in the pX459 plasmid (pSpCas9(BB)-2A-Puro; Addgene #48139), which encodes Cas9 and puromycin resistant genes. HeLa cells were transfected with these plasmids using Lipofectamine 2000 (Invitrogen). Cells were cultured in the presence of 1 $\mu$g/ml puromycin for 3 d, and then stable clones were established. Target gene deletion in the TICAM-1 locus was assessed by nucleotide sequencing. The absence of TICAM-1 was confirmed by Western blotting. Mouse primary NK, CD8[+] DCs, CD11c[+], and CD11b[+] cells were isolated from C57BL/6 mice using MACS system (Miltenyi Biotec). Primary mouse BMDC, bone-marrow-derived macrophages (BMM), and MEF were prepared as described previously (Oshiumi et al, 2011). Mouse hepatocytes are described previously (Suzuki et al, 2013). Preparation of mouse CTL was prepared as described previously (Azuma et al, 2016).

## Cell culture

HEK293FT, HeLa, RAW264.7, and L929 cells were cultured in DMEM (high-glucose) with 10% heat-inactivated FCS and 1% penicillin–streptomycin (P/S). These cells were incubated in a 5% CO$_2$ at 37°C. Primary MEFs were isolated from WT or *Ticam-1* KO mice at

percentages (G) of Gr-1[+] CD11b[+] neutrophils in the lung (PBS: WT n = 5, KO n = 5. IL-17A: WT n = 12, KO n = 11). **(H)** The expression level of Cxcl1 mRNA in the lung from mice treated as in panel F was determined by RT-qPCR (PBS: WT n = 6, KO n = 6. IL-17A: WT n = 8, KO n = 9). **(F, I)** The expression level of CXCL2 and IL-6 mRNA in the lung from mice treated as in panel (F) was determined through RT-qPCR. The results show ± SD means (PBS: WT n = 3, KO n = 3. IL-17A: WT n = 6, KO n = 4. *P < 0.05; one-way ANOVA). Data in (A, C, D, E) show mean ± SD pooled from three experiments. Data in (G, H) show mean ± SD pooled from two experiments. *P*-values calculated using *t* test except for (G) (Mann–Whitney test) and (H, I) (one-way ANOVA).

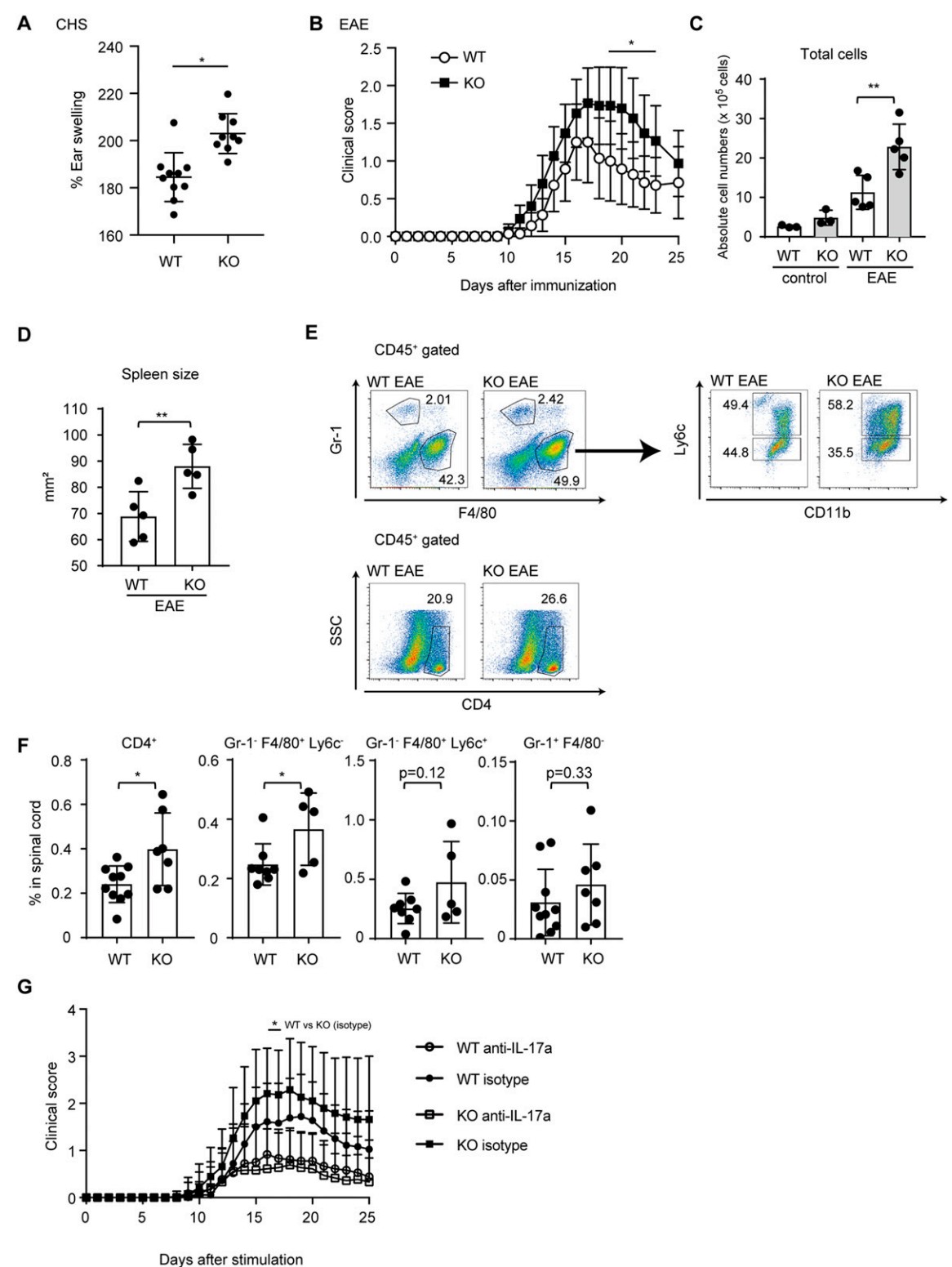

**Figure 6. TICAM-1 ameliorates experimental autoimmune encephalomyelitis (EAE) clinical scores.**
**(A)** % of ear swelling of WT and *Ticam-1* KO mice in contact hypersensitivity assay. Each dot represents one mouse (*$P < 0.05$, *t* test). **(B)** Clinical scores of EAE in WT (n = 14) and *Ticam-1* KO (n = 15) male mice with time. Mean ± SD are shown (*$P < 0.05$, two-way ANOVA). **(C)** Total cell counts in inguinal lymphoid nodes from naive WT, *Ticam-1* KO and WT, *Ticam-1* KO male mice analyzed on day 19 post-immunization with $MOG_{35-55}$ (control: WT n = 3, *Ticam-1* KO n = 3. EAE: WT n = 5, *Ticam-1* KO n = 5, *$P < 0.05$ one-way ANOVA). **(D)** Spleen size from WT, *Ticam-1* KO EAE male mice (n = 5). Spleens were analyzed on day 10 after immunization. Each dot represents one mouse (*$P < 0.05$, *t* test). **(E, F)** Representative plots (E) and percentages (F) of CD4+ T cells (WT n = 10, KO n = 7), Gr-1− F4/80+ Ly6c− macrophages (WT n = 8, KO n = 5), Gr-1− F4/80+ Ly6c+

embryonic day 14, and then cultured in DMEM (high-glucose) with 10% FCS and 1% P/S.

## Western blot analysis and immunoprecipitation

Cells were lysed with lysis buffer (10 mM Hepes, 50 mM NaCl, 1 mM EDTA, 10% glycerol, 1% Triton, 30 mM NaF, and 1 mM $Na_3VO_4$) in the presence of protease inhibitor cocktail tablet (Sigma-Aldrich). For immunoprecipitation, lysates were incubated with Protein G Sepharose beads and the antibody overnight. Samples were boiled and separated to Tris-glycine SDS–PAGE and transferred onto polyvinylidene fluoride (PVDF) membranes. The membranes were incubated with the primary antibody and HRP-conjugated secondary antibodies (GE Healthcare). Immunoblots were visualized using the Amersham ECL Prime Western blotting detection reagent (GE Healthcare) and Bio-Rad ChemiDoc touch. The band intensity was measured by Image Lab software.

## Plasmid

IL-17RA and Act1 plasmids were purchased from DNAFORM (IL-17RA: 4053746, Act1: 3634505). The IL-17RA-myc plasmid was reconstructed in the pcDNA-myc vector, and Act1-FLAG plasmid was reconstructed in the pEF-BOS FLAG vector using the In-Fusion cloning kit (TaKaRa). TICAM-1-HA plasmid was described previously (Oshiumi et al, 2003a). TICAM-1-TIR-HA and TICAM-1-ΔTIR-HA, Act1-SEFIR-FLAG, and Act1-ΔSEFIR-FLAG plasmids were constructed in the pEF-BOS vectors.

## Reporter gene assay

HeLa cells were co-transfected with NF-κB reporter (100 ng/ml) and phRL-TK (10 ng/well Promega) plasmids. The phRL-TK vector encodes *Renilla* luciferase. After 18 h, cells were stimulated with 50 ng/ml IL-17A for 6 h and lysed. Luciferase and *Renilla* luciferase activities were determined using the Dual-Luciferase Reporter Assay system (Promega). Luminescence was detected by using a luminometer (Cat. no. AB-2270; ATTO). Luciferase activity was normalized to *Renilla* luciferase activity.

## PLA

HeLa cells cultured on glass bottom plate were fixed with 4% PFA and were permeabilized with 0.3% Triton X-100 in PBS. PLA signals were detected by Duolink in situ PLA kit, according to the manufacturer's instruction (Sigma-Aldrich). In brief, HeLa cells were incubated with first antibody, and then secondary antibodies conjugated with the PLA probes were added. If the PLA probes are in proximity, the DNA synthesis is initiated. Finally, fluorescent probes that bind to the amplified DNA were added. We detected PLA signal intensity using microscopy software (BZ-X Analyzer) and normalized DAPI numbers.

## Quantitative real-time PCR

Total RNA was isolated using the TRI reagent (MOR), according to the manufacturer's instruction, and incubated for 5 min at 65°C. cDNA was obtained using the ReverTra Ace qPCR RT Master Mix with gDNA Remover (TOYOBO). qPCR was performed using the Power SYBR Green PCR Master Mix (Life Technologies) and Step-one Real-time PCR system (Life Technologies). Target mRNA expressions was determined by using the following primers: human (h) Il6 5'-AGG CAC TGG CAG AAA ACA AC-3' and 5'-TTT TCA CCA GGC AAG TCT CC-3'; hCXCL1 5'-TCC TGC ATC CCC CAT AGT TA-3' and 5'-CTT CAG GAA CAG CCA CCA GT-3'; hCXCL2 5'-CTC AAG AAT GGG CAG AAA GC-3' and 5'-AAA CACA TTA GGC GCA ATC C-3'; hIfnb 5'-TGG GAG GAT TCT GCA TTA CC-3' and 5'-CAG CAT CTG CTG GTT GAA-3'; hGapdh 5'-CAA TAT GAT TCC ACC CAT GG-3' and 5'-AAT GAG CCC CAG CCT TCT CC-3'. Mouse (m) Il6 5'-GTT CTC TGG GAA ATC GTG GA-3' and 5'-GGT ACT CCA GAA GAC CAG AGG A-3'; mCxcl1 5'-ATC AGC AGC TTG AAG GTG TTG-3' and 5'-GTC TGT CTT CTT TCT CCG TTA CTT-3'; mCxcl2 5'-ATG CCT GAA GAC CCT GCC AAG-3' and 5'-GGT CAG TTA GCC TTG CCT TTG-3'; mGapdh 5'-AAT GGT GAA GGT CGG TGT G-3' and 5'-GAA GAT GGT GAT GGG CTT CC-3'; mb-actin 5'-TCG TCA TCC ATG GCG AAC T-3' and 5'-TTT GCA GCT CCT TCG TTG C-3'. Expression of each gene was normalized to Gapdh expression using the comparative $2[-\Delta\Delta C_t]$ method. For measurement of *Cxcl1* RNA stability, cells were pretreated for 1 h with TNF-α (10 ng/ml), followed by the treatment with actinomycin D (5 µg/ml) alone or in combination with IL-17 stimulation (50 ng/ml) for various time. The decay of *Cxcl1* mRNA levels was calculated using the value of relative expression of *Cxcl1* mRNA at 0 min (starting time point) of each stimulatory condition as 100%.

## Flow cytometry

When the lung-infiltrating cells were analyzed, anti-CD45-APC-Cy7 (BioLegend) were intravenously injected (3 µg/mouse) before euthanasia to exclude the blood-circulating immune cells. Cells from lung or spinal cord tissues were prepared by enzymatic digestion with 2.5 mg/ml collagenase D (Sigma-Aldrich) and 0.1 mg/ml DNase I (Sigma-Aldrich) for 30 min at 37°C. For neutrophils, monocytes, macrophages, and CD4[+] T cell staining, the cells or tissues were stained with anti-Ly6c-AF488 (BioLegend), anti-CD45.2-PE (BioLegend), anti-CD11c-PE (BioLegend), anti-CD11b-PerCP (BioLegend), anti-Gr-1-APC (eBioscience), anti-CD4-APC (TONBO), and anti-F4/80-BV421 (BioLegend).

## Statistical analysis

Data were analyzed using unpaired *t* test when comparing two experimental groups. The Mann–Whitney test was used as a nonparametric alternative to *t* test. Multiple comparisons were performed using one-way ANOVA followed by the Tukey–Kramer

monocytes (WT n = 8, KO n = 5), and Gr-1[+] F4/80[−] neutrophils (WT n = 10, KO n = 7) in the spinal cord on day 19 after immunization. Cells were collected and then stained with indicated antibodies, and were subjected to flow cytometry analyses. Samples were extracted from WT and *Ticam-1* KO mice with EAE clinical score ≧ 1. **(G)** EAE scores of WT and *Ticam-1* KO male mice injected with 100 µg of anti-IL-17A (WT n = 18, KO n = 18) or isotype control antibody (WT n = 18, KO n = 19) on days −1, 2, 5, 8, 11, and 14 of EAE. EAE scores of WT and *Ticam-1* KO mice (isotype control) at each time point were compared for statistical analyses (*$P < 0.05$, two-way ANOVA). Data in (B) show mean ± SD pooled from two experiments. Data in (F) show mean ± SD pooled from three experiments. Data in (G) show mean ± SD pooled from four experiments. *P*-values calculated using *t* test except for (A, B, and F).

post hoc test. These analyses were performed using Prism ver.4.0. Error bars represent SD. *P*-values less than 0.05 were considered statistically significant.

## Data Availability

Data and details of this study are available from the corresponding authors upon request.

## Supplementary Information

## Acknowledgements

We thank Drs. Seya T (Hokkaido University), Matsumoto M (Aomori University), and all our laboratory members for helpful discussion, and Fukusima Y for technical assistance. This work was supported by Grants-in-Aid from Japan Society for the Promotion of Science (JSPS) KAKENHI Grant Number JPH03480 and 19H03480 and Japan Agency for Medical Research and Development.

### Author Contributions

Y Miyashita: data curation, formal analysis, validation, investigation, and writing—original draft, review, and editing.
T Kouwaki: data curation and investigation.
H Tsukamoto: data curation, supervision, investigation, and writing—original draft, review, and editing.
M Okamoto: data curation and investigation.
K Nakamura: conceptualization, supervision, and project administration.
H Oshiumi: conceptualization, resources, supervision, funding acquisition, project administration, and writing—original draft, review, and editing.

### Conflict of Interest Statement

The authors declare that they have no conflict of interest.

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
