## [Reviewer comments · Life Science Alliance]

Life Science Alliance

TICAM-1/TRIF associates with Act1 and suppresses IL-17 receptor-mediated inflammatory responses

Yusuke Miyashita, Takahisa Kouwaki, Hirotake Tsukamoto, Masaaki Okamoto, Kimitoshi Nakamura, and Hiroyuki Oshiumi
DOI: <https://doi.org/10.26508/lsa.202101181>

Corresponding author(s): Hiroyuki Oshiumi, Kumamoto University and Hirotake Tsukamoto, Kyoto University

Review Timeline:

Submission Date:	2021-07-30
Editorial Decision:	2021-08-24
Revision Received:	2021-10-29
Editorial Decision:	2021-11-11
Revision Received:	2021-11-12
Accepted:	2021-11-12

Scientific Editor: Novella Guidi

Transaction Report:

August 24, 2021

Re: Life Science Alliance manuscript #LSA-2021-01181

Prof. Hiroyuki Oshiumi
Kumamoto University
1-1-1 Honjo, Chuo-ku
Kumamoto 8608556
Japan

Dear Dr. Oshiumi,

Thank you for submitting your manuscript entitled "TICAM-1/TRIF associates with Act1 and suppresses IL-17 receptor-mediated inflammatory responses" to Life Science Alliance. The manuscript was assessed by expert reviewers, whose comments are appended to this letter. As you will note from the reviewers' comments below, both reviewers are quite positive about the work that in their view represents a straightforward and well written manuscript with data from both in-vitro and in-vivo experiments that appear technically sound, and most claims are supported by the data. They just point to few experiments to be conducted: Reviewer 1 argues that deletion mutants are needed to show that it is the TIR domain of TRIF that is responsible for ACT1 interaction and subsequent inhibition of the ACT1 / IL-17R interaction. Reviewer 2 requests that the authors analyze subcellular co-localization of TICAM-1, Act1, and IL-17A. Although you can't see the cross-commenting section, the two reviewers agreed that the following points could be overruled:

- Figure 3. The authors should show expression of TRIF in these panels. Firstly to show knockout and secondly to determine whether TRIF protein levels remain constant after IL-17A stimulation.
- In the Fig 4a, TRIF completely blocks IL-17RA and ACT1 interaction. To demonstrate that this is dependent on the TIR domain of TRIF, truncation mutants of TRIF lacking a TIR domain or containing only the TIR domain should be performed.
- ACT1 also induces a mRNA stability pathway. does TRIF affect this pathway? mRNA decay should be measured over time in response to IL-17A with cells treated with Actinomycin-D to inhibit new mRNA synthesis.

Besides these last points, we encourage you to submit a revised version of the manuscript back to LSA that responds to all the remaining reviewers' points.

Thank you for this interesting contribution to Life Science Alliance. We are looking forward to receiving your revised manuscript.

Sincerely,

- A letter addressing the reviewers' comments point by point.
- An editable version of the final text (.DOC or .DOCX) is needed for copyediting (no PDFs).

B. MANUSCRIPT ORGANIZATION AND FORMATTING:

Reviewer #1 (Comments to the Authors (Required)):

In the current manuscript, Miyashita et al. have focused on the similarity between the TIR domain of TICAM-1 and the SEFIR domains of IL-17RA/Act1 and found an actual physical interaction between TICAM-1 and Act1. They found that TICAM-1 expression interfered the IL-17R and Act1 interaction and TICAM-1 deficiency in cells enhanced the interaction, downstream signaling, and the chemokine gene induction following IL-17A stimulation. TICAM-1-deficient mice exhibited exacerbated CHS and EAE, in which IL-17 acts a pathogenic role, compared to wild-type mice. Overall, this paper represents a straight forward and well written manuscript with data from both in-vitro and in-vivo experiments that appear technically sound and most claims are supported by the data. Therefore, I have only few suggestions to improve this paper before acceptance for the publication.

Major points

- 1) In Figure 1: the authors should show that the interaction between TICAM-1 and Act1 are through their TIR and SEFIR domains by using deletion mutants.
- 2) In Figure 4a-c: Similarly, please examine if the inhibition of IL-17RA-Act1 interaction by TICAM-1 is dependent on its TIR domain.

Minor points

- 1) In Figure 5: It is unclear the time when the cells were prepared from peritoneal cavity and lungs after IL-17A administration. Please indicate it in the legend.
- 2) In Figure 6e, f : Please indicate in the legend whether the cells were prepared and analyzed after intravascular CD45 staining.

Reviewer #2 (Comments to the Authors (Required)):

1. Summary

Miyashita and colleagues present a new and novel role for the TLR adaptor molecule TRIF in IL-17 receptor signalling. The proposed work describes an interesting facet of the crosstalk between TLRs and cytokine receptors that fine-tunes the immune response to infection. It is interesting that TRIF, that promotes a type I Interferon response, would serve to dampen pro-inflammatory response against bacteria/fungi (i.e. Th17 driven responses). The authors describe how TRIF interacts with the IL-17RA adaptor Act1 and blocks its interaction with the IL-17 Receptor. In vivo TRIF KO animals, failing to control IL-17A-driven responses, show exacerbated responses in an EAE model and in response to IL-17A.

2. Suggested Additional Experiments

- for Figure 1 the interaction between ACT1 and TRIF should be complemented with PLA and confocal studies that were performed in figure 4. The cellular location of TRIF and ACT1 would be interesting in response to IL-17A. (in figure 1 the quantification of blots could be moved to supplementary if space is needed).

the reduction of Act1 and TRIF is more pronounced at 15 minutes (fig 1e). Why not try longer timepoints here. Strangely, the ACT1 and IL-17RA response is lost at 10 minutes (Fig 4C). Can the authors describe the discrepancy here? Does TRIF co-localise to the IL-17RA with ACT1?

- Figure 2. Although the effects of TRIF KO on IL-17A pathways in vitro are significant, they are quite "mild" and in figure 3 are at early timepoints. It would suggest that in wild-type cells IL-17A stimulation easily circumvents TRIF inhibition. It would greatly enhance the manuscript to understand this mechanism. What happens to TRIF upon IL-17A stimulation? Is the adaptor molecule, phosphorylated, ubiquitinated and/or degraded? Of course, ACT1 and other E3 ligases are involved in IL-17R signalling it would be interesting to assess TRIF ubiquitination and potential degradation in IL-17 signalling.

- Figure 3. The authors should show expression of TRIF in these panels. Firstly to show knockout and secondly to determine whether TRIF protein levels remain constant after IL-17A stimulation.

- the crosstalk between TRIF and the IL-17 pathway is interesting but what does it mean for the crosstalk between TLR3 and IL-17RA. Does Poly(I:C) inhibit IL-17A responses or vice versa?

- In Figure 4c and 4g (in WT and TRIF KO), the interaction between ACT1 and IL-17RA is lost at 15 minutes but in 4d it still remains. Can the authors discuss this discrepancy.

- In the Fig 4a, TRIF completely blocks IL-17RA and ACT1 interaction. To demonstrate that this is dependent on the TIR domain of TRIF, truncation mutants of TRIF lacking a TIR domain or containing only the TIR domain should be performed.

- ACT1 also induces a mRNA stability pathway. does TRIF affect this pathway? mRNA decay should be measured over time in response to IL-17A with cells treated with Actinomycin-D to inhibit new mRNA synthesis.

3. Minor comments:

- is it not surprising that TLR3 expression in BMDCs is very low (Fig 1C)? please comment.

- shorten description of Fig 1f

- When citing Notachkova (ref no. 25) please expand that SEFIR and TIR domains comprise part of a larger superfamily. After (Fig. 1b) also cite Novatchkova and Mellett papers as previously described as these groups previously also performed alignments.

- The first figure describes that TRIF cellular expression differs from TLR3, however, TRIF is also an adaptor molecule for TLR4 - this should be noted in the text as it is not very clear to the reader why the authors came to the conclusion that TRIF could be involved in IL-17RA signalling.

- several times the author describe that "TICAM-1 KO enhanced" - this is slightly confusing as TICAM-1 KO is also used to describe the cells or mice, please correct these incidents with TRIF knockout or deficiency. Similarly, TICAM-1 KO is used for human and murine, italics should be used for the human gene and lowercase letters and italics for the murine gene so it is clearer for readers.

- in Supplementary figure 1A, please add amino acid residue numbers at termini and at major domains.

- in discussion, in line 3 take out M to leave Mellett et al. On line 6 add "the" after resembles

- in last line of Figure 5 results section add "regulating" before "IL-17A-mediated"

- in results section for fig 2, add "knock-down" after TRIF siRNA

- in results section for figure 4, HEK293 T cells has a typo.

- change to "enhanced in TICAM-1 KO mice" before (Fig. 6a)

- please add page numbers for easy review

Point-by-point response

Reviewer #1 comment:

In the current manuscript, Miyashita et al. have focused on the similarity between the TIR domain of TICAM-1 and the SEFIR domains of IL-17RA/Act1 and found an actual physical interaction between TICAM-1 and Act1. They found that TICAM-1 expression interfered the IL-17R and Act1 interaction and TICAM-1 deficiency in cells enhanced the interaction, downstream signaling, and the chemokine gene induction following IL-17A stimulation. TICAM-1-deficient mice exhibited exacerbated CHS and EAE, in which IL-17 acts a pathogenic role, compared to wild-type mice.

Overall, this paper represents a straight forward and well written manuscript with data from both in-vitro and in-vivo experiments that appear technically sound and most claims are supported by the data. Therefore, I have only few suggestions to improve this paper before acceptance for the publication.

Response:

We are grateful to you for helpful comments and suggestions. We have addressed all of the reviewer comments and revised our manuscript throughout as described below.

Major point 1:

In Figure1: the authors should show that the interaction between TICAM-1 and Act1 are through their TIR and SEFIR domains by using deletion mutants.

Response:

Thank you for your suggestion. We have added new data. First, we found that TICAM-1 TIR domain was co-immunoprecipitated with Act1, and that the deletion of the TIR domain abolished the interaction (new Fig 1E). Second, we found that TICAM-1 was not co-immunoprecipitated with SEFIR-domain-deleted Act1 (new Fig 1F). These data indicate that the interaction between TICAM-1 and Act1 are through their TIR and SEFIR domains.

Major point 2:

In Figure 4a-c: Similarly, please examine if the inhibition of IL-17RA-Act1 interaction by TICAM-1 is dependent on its TIR domain.

Response:

Thank you for your helpful comment. We have investigated whether the inhibition by TICAM-1 is dependent on its TIR domain. New Figs 4D and S4A showed that the expression of TIR domain of TICAM-1 could inhibit the interaction between IL-17RA and Act1. These data indicate that the inhibition is dependent on the TIR domain of TICAM-1.

Minor point 1:

In Figure 5: It is unclear the time when the cells were prepared from peritoneal cavity and lungs after IL-17A administration. Please indicate it in the legend.

Response:

Thank you for helpful comment. We have indicated the time when cells were prepared from peritoneal cavity and lungs after IL-17A administration (Figure legends of Fig 5F and G (line 752)).

Minor point 2:

In Figure 6e, f: Please indicate in the legend whether the cells were prepared and analyzed after intravascular CD45 staining.

Response:

Thank you for your suggestion. In Figs 6E and F, cells were prepared at 19 days post EAE induction, and stained with CD45 after cells were collected. This information was described in our revised manuscript (lines 249–250 and 787-788).

Reviewer #2 comment:

Miyashita and colleagues present a new and novel role for the TLR adaptor molecule TRIF in IL-17 receptor signalling. The proposed work describes an interesting facet of the crosstalk between TLRs and cytokine receptors that fine-tunes the immune response to

infection. It is interesting that TRIF, that promotes a type I Interferon response, would serve to dampen pro-inflammatory response against bacteria/fungi (i.e. Th17 driven responses). The authors describe how TRIF interacts with the IL-17RA adaptor Act1 and blocks its interaction with the IL-17 Receptor. In vivo TRIF KO animals, failing to control IL-17A-driven responses, show exacerbated responses in an EAE model and in response to IL-17A.

Response:

We are grateful to you for helpful comments and suggestions. We have addressed all of the reviewer comments and revised our manuscript throughout as described below.

Comment 1:

*- for Figure 1 the interaction between ACT1 and TRIF should be complemented with PLA and confocal studies that were performed in figure 4. The cellular location of TRIF and ACT1 would be interesting in response to IL-17A. (in figure 1 the quantification of blots could be moved to supplementary if space is needed).
the reduction of Act1 and TRIF is more pronounced at 15 minutes (fig 1e). Why not try longer timepoints here. Strangely, the ACT1 and IL-17RA response is lost at 10 minutes (Fig 4C). Can the authors describe the discrepancy here? Does TRIF co-localise to the IL-17RA with ACT1?*

Response:

Thank you for your helpful comment. First, we have performed the PLA to observe the association between Act1 and TICAM-1. New Fig S1D showed that exogenously expressed FLAG-tagged Act1 was co-localized with HA-tagged TICAM-1 in the cytoplasm. Second, we detected the colocalization between endogenous Act1 and TICAM-1, and found that the number of colocalization signals (PLA signals) were reduced by IL-17A stimulation (new Fig 1J). We have moved old Fig 1F to Fig S1C, according to the reviewer suggestion, because there was no sufficient space for new Fig 1J (as well as new Fig 1E and 1F).

Third, as the reviewer pointed out, the kinetics of the reduction of TICAM-1 with Act1 were a little bit different between Fig 1E and 4C. According to reviewer suggestion, we have observed the interaction at later time point and have replaced with old Fig 1E

with new Fig 1G. These new and previous data showed that Act1 bound to IL-17RA immediately after IL-17A stimulation, whereas TICAM-1 dissociated from Act 1 at later time points. One possibility is that dissociation of TICAM-1 is not a prerequisite for the interaction between Act1 and IL-17RA, and the dissociation might modulate the binding stability. There are several other possibilities. For instance, TICAM-1 might directly disrupt the interaction between IL-17RA and ACT1, or it might sequester ACT1, which keeps ACT1 away from IL-17RA. We have discussed this point in our revised manuscript in Discussion (lines 277–288).

Comment 2:

- Figure 2. Although the effects of TRIF KO on IL-17A pathways in vitro are significant, they are quite "mild" and in figure 3 are at early timepoints. It would suggest that in wild-type cells IL-17A stimulation easily circumvents TRIF inhibition. It would greatly enhance the manuscript to understand this mechanism. What happens to TRIF upon IL-17A stimulation? Is the adaptor molecule, phosphorylated, ubiquitinated and/or degraded? Of course, ACT1 and other E3 ligases are involved in IL-17R signalling it would be interesting to assess TRIF ubiquitination and potential degradation in IL-17 signalling.

Response:

Thank you for pointing out an interesting point. We have investigated the ubiquitination and phosphorylation TICAM-1 (TRIF) upon IL-17A stimulation (new Fig S3A and S3B), but we could not detect any ubiquitination or phosphorylation of TICAM-1 after IL-17A stimulation as far as we tested. Additionally, we have investigated the TICAM-1 protein levels (new Fig 3A, B, and D) and found that TICAM-1 was not degraded after IL-17A stimulation. But, we do not exclude the possibility that TICAM-1 harbors other post-translational modification, such as modifications with ubiquitin-like proteins. We have discussed this point in our revised manuscript (lines 285-286).

Comment 3:

- Figure 3. The authors should show expression of TRIF in these panels. Firstly to show knockout and secondly to determine whether TRIF protein levels remain constant after IL-17A stimulation.

Response:

Thank you for your helpful comment. We have observed the TICAM-1 protein levels after IL-17A stimulation in new Fig 3. New Figs 3A, B, and D showed that the TICAM-1 protein was not detectable in TICAM-1 KO cells and that the TICAM-1 protein levels were barely affected by IL-17A stimulation.

Comment 4:

- the crosstalk between TRIF and the IL-17 pathway is interesting but what does it mean for the crosstalk between TLR3 and IL-17RA. Does Poly(I:C) inhibit IL-17A responses or vice versa?

Response:

Thank you for your helpful comment. Since previous studies have already shown that poly I:C does not inhibit IL-17A response or vice versa, we prefer the interpretation that the TICAM-1 expression levels would determine the balance between IL-17A and TLR3 signaling pathways. Indeed, knockout of TICAM-1 augmented IL-17A-mediated cytokine expression, and overexpression of TICAM-1 inhibited the interaction of IL-17RA with Act1. We have discussed this point in our revised manuscript (lines 264-266).

Comment 5:

- In Figure 4c and 4g (in WT and TRIF KO), the interaction between ACT1 and IL-17RA is lost at 15 minutes but in 4d it still remains. Can the authors discuss this discrepancy.

Response:

Thank you for your helpful comments. In old Fig 4C, cells were transfected using lipofectamine reagents. Thus, it is possible that the differences in the experimental conditions might affect the sensitivity of cells to IL-17A. We have discussed the differences in the kinetics of the binding among TICAM-1, Act1, and IL-17RA (lines 277–288).

Comment 6:

- In the Fig 4a, TRIF completely blocks IL-17RA and ACT1 interaction. To demonstrate that this is dependent on the TIR domain of TRIF, truncation mutants of TRIF lacking a TIR domain or containing only the TIR domain should be performed.

Response:

Thank you for helpful suggestion. We have investigated whether the expression of the TIR domain of TICAM-1 inhibits the interaction between IL-17RA and Act1. New Fig 4D and new Fig S4A showed that the TIR domain was sufficient to inhibit the interaction.

Comment 7:

- ACT1 also induces a mRNA stability pathway. does TRIF affect this pathway? mRNA decay should be measured over time in response to IL-17A with cells treated with Actinomycin-D to inhibit new mRNA synthesis.

Response:

Thank you for helpful suggestion. We have investigated whether TICAM-1 affected the Act1-mediated mRNA stability. WT and TICAM-1 KO cells were treated with actinomycin-D in the presence and absence of IL-17A, and the decay of CXCL1 mRNA was measured by RT-qPCR. New Fig S2B showed that there were no differences in the degradation rates of CXCL1 mRNA between WT and TICAM-1 KO cells. This data weakened the possibility that TICAM-1 affected the Act1-mediated mRNA stability.

Minor comment 1:

- is it not surprising that TLR3 expression in BMDCs is very low (Fig 1C)? please comment.

Response:

Previous studies have shown that TLR3 is highly expressed in CD8+ DCs but not in other types of DCs, such as CD4+ DCs, and DN DCs. We have mentioned this point in our revised manuscript (lines 106–107).

Minor comment 2:

- shorten description of Fig 1f

Response:

Thank you for helpful comment. We have improved the description (lines 117–118).

Minor comment 3:

- *When citing Notachkova (ref no. 25) please expand that SEFIR and TIR domains comprise part of a larger superfamily. After (Fig. 1b) also cite Novatchkova and Mellett papers as previously described as these groups previously also performed allignments.*

Response:

Thank you for helpful comments. We have improved our explanation, according to your suggestions (lines 89–90), and have cited both Novatchkova and Mellett group papers after Fig 1B (lines 103–104).

Minor comment 4:

- *The first figure describes that TRIF cellular expression differs from TLR3, however, TRIF is also an adaptor molecule for TLR4 - this should be noted in the text as it is not very clear to the reader why the authors came to the conclusion that TRIF could be involved in IL-17RA signalling.*

Response:

Thank you for helpful comment. We have mentioned that TICAM-1 is an adaptor of TLR4 in our revised manuscript (lines 64–66, 111)

Minor comment 5:

- *several times the author describe that "TICAM-1 KO enhanced" - this is slightly confusing as TICAM-1 KO is also used to describe the cells or mice, please correct these incidents with TRIF knockout or deficiency. Similarly, TICAM-1 KO is used for human and murine, italics should be used for the human gene and lowercase letters and italics for the murine gene so it is clearer for readers.*

Response:

Thank you for helpful comment. We have rephrased “KO” with “knockout” to avoid any confusion throughout, and we have used uppercase and lowercase letters and italics for the human and murine genes.

Minor comment 6:

- *in Supplementary figure 1A, please add amino acid residue numbers at termini and at major domains.*

Response

Thank you for helpful suggestions. We have added the aa residue numbers at each major domain in revised Fig S1A.

Minor comment 7

- *in discussion, in line 3 take out M to leave Mellett et al. On line 6 add "the" after resembles*

Response:

Thank you for helpful comment. We have taken out M to leave Mellet et al. and have added “the” after resembles (lines 262, 267).

Minor comment 8

- *in last line of Figure 5 results section add "regulating" before "IL-17A-mediated"*

Response:

Thank you for helpful comment. We have added “regulating”, according to the suggestion (line 227).

Minor comment 9

- *in results section for fig 2, add "knock-down" after TRIF siRNA*

Response:

Thank you for your helpful comment. We have added “knockdown” after siRNA (lines 137, 145).

Minor comment 10

- in results section for figure 4, HEK293 T cells has a typo.

Response:

Thank you for your comment. But, in Figure 4, we used HEK293FT cells instead of HEK293T cells.

Minor comment 11

- change to "enhanced in TICAM-1 KO mice" before (Fig. 6a)

Response:

Thank you for helpful comment. We have changed the sentence according to the suggestion (line 238).

Minor comment 12

- please add page numbers for easy review

Response:

We regret this inconvenience. We have added the page numbers and line numbers in the revised manuscript.

November 11, 2021

RE: Life Science Alliance Manuscript #LSA-2021-01181R

Prof. Hiroyuki Oshiumi
Kumamoto University
1-1-1 Honjo, Chuo-ku
Kumamoto 8608556
Japan

Dear Dr. Oshiumi,

Thank you for submitting your revised manuscript entitled "TICAM-1/TRIF associates with Act1 and suppresses IL-17 receptor-mediated inflammatory responses". We would be happy to publish your paper in Life Science Alliance pending final revisions necessary to meet our formatting guidelines.

- please add the Twitter handle of your host institute/organization as well as your own or/and one of the authors in our system
- please be sure that all authors are listed in the authors' contribution section
- please add your main and supplementary figure legends to the main manuscript text after the references section
- all figure legends should only appear in the main manuscript file
- figure S4 has only one panel, please revise its legend and remove the label A from the actual figure. Correct the callout in the manuscript text accordingly
- please indicate scale bar size in Legends for figures 1J and S1D
- from Reviewer 2: One misunderstanding: ticam-1 KO mice in italics should still have an uppercase "T", i.e. Ticam-1 (apologies for the misunderstanding)

A. FINAL FILES:

B. MANUSCRIPT ORGANIZATION AND FORMATTING:

Sincerely,

Reviewer #1 (Comments to the Authors (Required)):

The authors have addressed all the points raised by this reviewer and revised the article appropriately. Thank you for the opportunity to review this interesting work.

Reviewer #2 (Comments to the Authors (Required)):

1. Miyashita et al. have demonstrated that the Toll-like Receptor adaptor molecule, TICAM-1 regulates IL-17 signalling by directly interacting between the TIR domain of TICAM-1 and the SEFIR domains of Act1. TICAM-1 dampens IL-17-induced inflammation and this has physiological impact as TICAM-1-deficient mice exhibited exacerbated CHS and EAE, in which IL-17 acts a pathogenic role, compared to wild-type mice. Overall, this work a straightforward and well written manuscript with supportive data from both in-vitro and in-vivo experiments that put forward evidence of the cross-talk between Toll-like Receptors and IL-17 signalling in fine-tuning a tailored inflammatory response to pathogens.

2. The conclusions are supported by the data.

3. The authors have answered all my questions. One misunderstanding: *ticam-1* KO mice in italics should still have an uppercase "T", i.e. *Ticam-1* (apologies for the misunderstanding).

Point-by-point response

Reviewer #1 (Comments to the Authors (Required)):

The authors have addressed all the points raised by this reviewer and revised the article appropriately.

Thank you for the opportunity to review this interesting work.

Response:

We are grateful to you for reviewing our manuscript.

Reviewer #2 (Comments to the Authors (Required)):

Comment 1. Miyashita et al. have demonstrated that the Toll-like Receptor adaptor molecule, TICAM-1 regulates IL-17 signalling by directly interacting between the TIR domain of TICAM-1 and the SEFIR domains of Act1. TICAM-1 dampens IL-17-induced inflammation and this has physiological impact as TICAM-1-deficient mice exhibited exacerbated CHS and EAE, in which IL-17 acts a pathogenic role, compared to wild-type mice. Overall, this work a straightforward and well written manuscript with supportive data from both in-vitro and in-vivo experiments that put forward evidence of the cross-talk between Toll-like Receptors and IL-17 signalling in fine-tuning a tailored inflammatory response to pathogens.

Response:

We are grateful to you for reviewing our manuscript.

Comment 2. The conclusions are supported by the data.

Comment 3. The authors have answered all my questions. One misunderstanding: ticam-1 KO mice in italics should still have an uppercase "T", i.e. Ticam-1 (apologies for the misunderstanding).

Response:

Thank you for your helpful comment. We have corrected our mistakes. We have replaced all "*ticam-1*" with "*Ticam-1*" throughout our manuscript.

November 12, 2021

RE: Life Science Alliance Manuscript #LSA-2021-01181RR

Prof. Hiroyuki Oshiumi
Kumamoto University
1-1-1 Honjo, Chuo-ku
Kumamoto 8608556
Japan

Dear Dr. Oshiumi,

Thank you for submitting your Research Article entitled "TICAM-1/TRIF associates with Act1 and suppresses IL-17 receptor-mediated inflammatory responses". It is a pleasure to let you know that your manuscript is now accepted for publication in Life Science Alliance. Congratulations on this interesting work.

DISTRIBUTION OF MATERIALS:

Again, congratulations on a very nice paper. I hope you found the review process to be constructive and are pleased with how the manuscript was handled editorially. We look forward to future exciting submissions from your lab.

Sincerely,
